# Glaucoma Detection through a Novel Hyperspectral Imaging Band Selection and Vision Transformer Integration

**DOI:** 10.3390/diagnostics14121285

**Published:** 2024-06-18

**Authors:** Ching-Yu Wang, Hong-Thai Nguyen, Wen-Shuang Fan, Jiann-Hwa Lue, Penchun Saenprasarn, Meei-Maan Chen, Shuan-Yu Huang, Fen-Chi Lin, Hsiang-Chen Wang

**Affiliations:** 1Department of Ophthalmology, Dalin Tzu Chi Hospital, Buddhist Tzu Chi Medical Foundation, Chiayi 62247, Taiwan; s19001052@gmail.com (C.-Y.W.); wsfan@tzuchi.com.tw (W.-S.F.); 2Department of Mechanical Engineering, National Chung Cheng University, Chiayi 62102, Taiwan; d09420004@ccu.edu.tw; 3Department of Optometry, Central Taiwan University of Science and Technology, No. 666, Buzih Road, Taichung City 406053, Taiwan; 108362@ctust.edu.tw (J.-H.L.); 108695@ctust.edu.tw (S.-Y.H.); 4School of Nursing, Shinawatra University, 99 Moo 10, Bangtoey, Samkhok, Pathum Thani 12160, Thailand; penchun_sa@yahoo.com; 5Center for Innovative Research on Aging Society (CIRAS), National Chung Cheng University, 168, University Rd., Min Hsiung, Chiayi 62102, Taiwan; laicmm@ccu.edu.tw; 6Department of Ophthalmology, Kaohsiung Armed Forces General Hospital, 2, Zhongzheng 1st. Rd., Kaohsiung City 80284, Taiwan; 7Hitspectra Intelligent Technology Co., Ltd., Kaohsiung City 80661, Taiwan

**Keywords:** glaucoma detection, hyperspectral imaging, vision transformer

## Abstract

Conventional diagnostic methods for glaucoma primarily rely on non-dynamic fundus images and often analyze features such as the optic cup-to-disc ratio and abnormalities in specific retinal locations like the macula and fovea. However, hyperspectral imaging techniques focus on detecting alterations in oxygen saturation within retinal vessels, offering a potentially more comprehensive approach to diagnosis. This study explores the diagnostic potential of hyperspectral imaging for glaucoma by introducing a novel hyperspectral imaging conversion technique. Digital fundus images are transformed into hyperspectral representations, allowing for a detailed analysis of spectral variations. Spectral regions exhibiting differences are identified through spectral analysis, and images are reconstructed from these specific regions. The Vision Transformer (ViT) algorithm is then employed for classification and comparison across selected spectral bands. Fundus images are used to identify differences in lesions, utilizing a dataset of 1291 images. This study evaluates the classification performance of models using various spectral bands, revealing that the 610–780 nm band outperforms others with an accuracy, precision, recall, F1-score, and AUC-ROC all approximately at 0.9007, indicating its superior effectiveness for the task. The RGB model also shows strong performance, while other bands exhibit lower recall and overall metrics. This research highlights the disparities between machine learning algorithms and traditional clinical approaches in fundus image analysis. The findings suggest that hyperspectral imaging, coupled with advanced computational techniques such as the ViT algorithm, could significantly enhance glaucoma diagnosis. This understanding offers insights into the potential transformation of glaucoma diagnostics through the integration of hyperspectral imaging and innovative computational methodologies.

## 1. Introduction

The physiological structure of the normal eye is similar to a spherical form, which is approximately 2 cm in diameter. It encapsulates a vital component known as aqueous humor, which is a watery substance that undergoes regular circulation to sustain various ocular elements. The dynamic equilibrium of aqueous humor circulation is pivotal, given that it necessitates fluid efflux through minute openings at the anterior part of the eye to re-enter the body. When these apertures become constricted or obstructed, a consequential fluid backlog ensues, which precipitates elevated intraocular pressure and thus instigates damage to the optic nerve. This condition is recognized as glaucoma, which may progress to irreversible blindness if left unattended. Glaucoma materializes when heightened intraocular pressure adversely impacts the optic nerve, which establishes a vital link between the eye and the brain. This malady is characterized by its potential to culminate in vision loss or, ultimately, blindness [1,2,3]. Therefore, early detection and intervention are imperative. Given its latent manifestation in the initial stages, glaucoma poses a challenge for timely recognition, considering the absence of symptomatic indicators. Nevertheless, prompt identification and judicious therapeutic intervention can halt the progression of the disease, which mitigates severe vision impairment and visual field loss. Although the exact cause of glaucoma remains unknown, it is closely linked to increased pressure within the eye and reduced circulation to the optic nerve. The onset of glaucoma typically occurs in individuals aged 40 or older, with a predilection for manifestation in the seventh and eighth decades of life [4]. This condition is particularly observed among females experiencing heightened anxiety and stress [4,5,6]. In addition, a hereditary predisposition is discernible, given that individuals with familial ties to glaucoma-affected individuals face an increased susceptibility—five to six times more likely—of succumbing to the ailment [7,8,9]. The insidious nature of glaucoma is further compounded by its asymptomatic presentation and the potential for misdiagnosis, especially among elderly individuals who may conflate it with presbyopia. Consequently, detection is frequently belated, with patients seeking medical attention only when visual impairment is advanced. This stage signifies irreversible damage to the optic nerve.

Several diagnostic methods are employed in the detection of glaucoma. Visual acuity, which is assessed through eye chart testing, serves to measure vision levels at varying distances [10,11,12,13,14]. Ophthalmoscopy involves the meticulous examination of the retina and optic nerve using a specialized magnifying lens by a medical professional to discern potential eye issues. Eye pressure measurement, which is accomplished through a dedicated instrument, aids in the identification of glaucoma by assessing intraocular pressure [15,16]. Retinal nerve fiber layers (RNFLs) analysis, corneal testing for the measurement of corneal thickness, and optical coherence tomography (OCT) for detecting damage to the optic nerve head (ONH) and RNFLs are additional diagnostic approaches [17,18,19,20,21,22]. Among these methods, ophthalmoscopy and OCT analysis of ocular nerves are notably regarded as highly reliable diagnostic modalities [13]. They are primarily employed in the diagnosis of glaucoma.

The fundus imaging method involves the analysis of images captured by a fundus camera. Evaluation of fundus images involves detecting irregularities in the structure of the ONH. Fundus imaging is progressively assuming a requisite role in glaucoma screening within ophthalmological facilities due to its cost effectiveness [23,24,25,26]. Diagnosing glaucoma through fundus images primarily depends on the ophthalmologist’s skill, with a significant focus on assessing the ONH. Four primary alterations in retinal nerve structures associated with glaucoma can be discerned: ONH cupping, neuro-retinal rim thinning, RNFL defects, and peripapillary atrophy. Furthermore, the analysis of the optic disc size contributes significantly to glaucoma grading [27,28]. Larger discs often exhibit more prominent cups, which potentially leads to a semblance of glaucoma and the risk of overestimation. Meanwhile, in smaller discs, a minor cup may indicate glaucoma, which poses the risk of underestimation. The World Glaucoma Association emphasizes the importance of optic disc clinical examination for glaucoma diagnosis, highlighting the assessment of the retinal nerve rim’s color and shape as crucial for differentiating between glaucomatous and non-glaucomatous neurological diseases [29,30,31,32,33]. A pallor of the neuroretinal rim is highly specific (94%) for non-glaucomatous optic neuropathy diagnosis. Similarly, both focal and diffuse thinning of the neuroretinal rim indicate glaucoma with an 87% specificity. Additionally, peripapillary atrophy is considered an indicative sign of glaucoma [34].

## 2. Literature Review

The integration of fundus imaging with artificial intelligence has been instrumental in enhancing glaucoma detection. Artificial intelligence approaches frequently employ image processing algorithms in conjunction with deep learning models for the classification of normal and glaucomatous eyes. In addition, segmentation algorithms are employed to delineate the positions of optic discs and cups, which facilitate the analysis of size relationships between them and the identification of potential abnormalities [27,28,35]. However, existing detection methods commonly depend on traditional convolutional neural network (CNN) structures, which necessitate substantial and uniform datasets. The aforementioned methods often incorporate external datasets in conjunction with proprietary data for training and validation purposes.

In addition to the emergence of advanced observational technologies, hyperspectral imaging technology is crucial in the exploration of glaucoma detection within the spectral domain. Approaches employing hyperspectral imaging technology predominantly utilize a fundus hyperspectral camera. Mordant et al. [36] proposed a method to evaluate retinal vascular oxygen saturation in normal eyes versus eyes treated for asymmetrical primary open-angle glaucoma (POAG) using a hyperspectral fundus camera. Utilizing noninvasive imaging and algorithmic analysis, the results revealed significantly higher retinal venular oxygen saturations in less- and more-advanced-treated POAG eyes compared with normal eyes. This condition suggests a potential indication of reduced oxygen consumption in the inner retinal tissues of advanced-treated POAG eyes. Our research group also has approaches to diagnosing eye diseases using hyperspectral imaging. In 2020, we presented a methodology employing hyperspectral imaging on ophthalmoscope images for diabetic retinopathy (DR) stage identification. By analyzing average reflectance spectra and utilizing principal component analysis (PCA), the method accurately categorizes DR stages. It demonstrates high sensitivity and accuracy in diagnosing normal, background DR, pre-proliferative DR, and proliferative DR based on oxygen saturation patterns in retinal vessels [37]. In 2023, we investigated hydroxychloroquine-induced retinopathy using color fundus images and employing hyperspectral conversion technology and deep learning models for lesion detection. The results demonstrate high overall accuracy, with EfficientNet achieving 94% accuracy for original images and 97% accuracy for hyperspectral images. This method offers a potential advanced diagnostic tool for identifying imperceptible lesions caused by hydroxychloroquine [38].

Diagnostic methods for glaucoma, whether employed in clinical diagnosis or integrated into computer-aided diagnosis, frequently rely on the analysis of fundus images. These approaches commonly draw upon identification features outlined in reference documents, including parameters such as the ratio between the size of the optic cup and the optic disc. Furthermore, abnormalities at specific locations such as the macula and fovea are scrutinized. By contrast, techniques utilizing hyperspectral imaging predominantly focus on investigating alterations in oxygen saturation within retinal vessels [36,39], or examine the transparency of the neural retinal rim through a reflectance signal [29,30,31,40,41,42]. The majority of fundus images utilized in this context are non-dynamic, which facilitates the application of hyperspectral imaging without the need to address camera kinematics. Instead, the emphasis is directed toward error correction control and chromatic adaptation. 

In this study, our aim is to provide a distinct perspective on the diagnostic approach to glaucoma, particularly the spectral domain methodology by hyperspectral imaging (HSI) and the integration of spatial information identification facilitated by the Vision Transformer (ViT) algorithm. We seek to discern the disparities between the machine learning algorithm’s perspective and the conventional clinical approach employed in fundus image analysis. To achieve this purpose, we introduce a hyperspectral imaging conversion technique designed to transform digital fundus images into hyperspectral representations as shown in Figure 1. We scrutinize and identify spectral regions exhibiting variations by employing spectral analysis. Subsequently, image reconstruction from these specific spectral regions is performed, which yields hyperspectral images corresponding to the investigated spectra. An examination of the reflectance signals in the optic disc and optic cup areas reveals information about the color of the neuroretinal rim. Simultaneously, the ViT model is utilized to harness spatial information, such as the optic cup to optic disc ratio (c/d ratio) and the spatial correlation of features surrounding the neuroretinal rim. The integration of spectral domain analysis through HSI with spatial information processing by ViT offers a novel perspective in the analysis of fundus images.

In this study, our contributions are as follows:We propose a hyperspectral imaging method combined with a deep learning model to help identify glaucoma based on fundus images;Band selection is based on computational methods to select the most optimal band range, thereby minimizing computational resources and improving prediction accuracy;The survey compares the prediction results with other state-of-the-art models, demonstrating that our model shows superiority.

## 3. Materials and Methods

### 3.1. Data Preparation

A dataset composed of 1291 fundus images was obtained from the Department of Ophthalmology at Dalin Hospital in Taiwan. The images were categorized into two groups based on their pathology: glaucoma and normal. Disease classification was conducted through diagnostic analyses carried out by ophthalmological specialists, which included the examination of OCT images and the assessment of the RNFL. The importance of integrating visual field assessments with evaluations of the nerve fiber layer is underscored, given that reliance on visual fields alone lacks sufficient reliability. Objective evidence of early glaucoma damage is primarily located in the optic disc and RNFL. The dataset encompasses a total of 1291 RNFL data points, comprising OCT images, thickness graphs as part of the OCT analysis report, and fundus images. Additionally, demographic data, including age, gender, and underlying health conditions, were collected for each participant. The process for examining glaucoma consists of a clinical diagnosis phase, an OCT RNFL report analysis phase, and a subsequent stage for monitoring the progression of the RNFL.

The RNFL data were obtained using a NIDEK RS-3000 Advance OCT (NIDEK Co., Ltd, Gamagori, Japan), an OCT system equipped with a scanning laser ophthalmoscope (SLO), designed for a thorough evaluation of the retina and choroid. The RS-3000 Advance offers exceptional detail of retinal and choroidal microstructures, thereby aiding in clinical diagnoses. Its integration of a retinal camera with OCT technology not only conserves space but also furnishes comprehensive diagnostic information. Notable features of the RS-3000 Advance include its capability to provide a holistic solution for retina and glaucoma analysis, precise image capture utilizing a SLO-based eye tracer, adjustable OCT sensitivity for the acquisition of B-scan images through media opacities, tracing HD for precise averaging of up to 120 images, extensive glaucoma analysis with a wide-area standard database measuring 9 × 9 mm, and high-resolution AngioScan OCT-Angiography imaging.

The RNFL report is depicted in Figure 2, which illustrates the OCT images in Figure 2a. Accompanying this, Figure 2b displays a red-free fundus photography image that highlights RNFL loss and indicates the locations of the scanned quadrants. Figure 2c presents the TSNIT Thickness Graph, where TSNIT stands for Temporal, Superior, Nasal, Inferior, and Temporal, delineating the color map area that corresponds to the scan layer thickness distribution database for the Temporal, Superior, Nasal, Inferior, and Temporal scanned areas, respectively. Within this graph, the blue area signifies the thickness typical of individuals diagnosed as normal with regard to glaucoma, the red area denotes those diagnosed with glaucomatous syndrome, and the yellow area indicates a suspected predisposition towards glaucoma. The line within the graph represents the patient’s measured values, while bar graphs associated with specific viewing areas depict particular thickness values. As evidenced in the sample report, the scan and line scan areas align closely with locations identified as having glaucoma. Nevertheless, a definitive diagnosis of glaucoma necessitates ongoing monitoring through the RNFL. This approach underscores the importance of comprehensive analysis and regular monitoring in the accurate diagnosis and management of glaucoma, facilitating early detection and intervention.

Through the analysis of OCT reports, a total of 1291 fundus images were curated and subsequently classified into two distinct groups based on the presence or absence of glaucoma. The classification yielded 638 images categorized under the glaucoma group, illustrating various stages and manifestations of the disease. Conversely, the remaining 653 images were classified as representing eyeballs without glaucoma, serving as a control group to facilitate comparative analysis and enhance the understanding of glaucomatous changes. This classification not only underscores the prevalence of glaucoma in the studied population but also provides a substantial dataset for further research and analysis aimed at improving diagnostic criteria and treatment strategies for glaucoma.

### 3.2. Optic Disc Localization

The critical information required to analyze the reflectance signal predominantly resides at the transparent level of the neuro-retinal rim area, which is demarcated by the boundary between the optic cup and optic disc, as illustrated in Figure 3. In this research, the process of localization involves the input of RGB retinal images and the output of region of interest (ROI) images. To ensure coverage of the largest optic disc (OD), the dimension of the ROI was standardized at 450 × 450 pixels.

The ResNet50 model, renowned for its deep CNN architecture, was employed as the foundational model for this study, excluding the top layer. This exclusion facilitates automatic cropping of regions encapsulating the optic discs and optic cups. ResNet50’s efficiency in various computer vision tasks, attributed to its capability to discern complex hierarchical image features, makes it a prime candidate for transfer learning in this context. The pre-trained layers of the ResNet model are frozen to preserve their acquired features, allowing for fine-tuning specific to this application. A novel model is then constructed atop the ResNet framework, incorporating additional layers where global average pooling diminishes the spatial dimensions of the feature maps to a singular value per channel. Subsequently, dense layers interpret these condensed feature vectors, determining the bounding box coordinates. The compilation of this model utilizes the Adam optimizer alongside a mean square error loss function, with a minor learning rate, suggested at 0.0001, to maintain the integrity of the pre-trained weights during fine-tuning.

### 3.3. Vision Transformer and Experimental Setup

The Vision Transformer (ViT) [43] embeds an input image into a sequence of patches, which is similar to a sequence of a word. Various studies have been made to apply the transformer structure to computer vision tasks. However, its application in medical image processing is still modest compared with that in other fields. Several medical imaging research studies have utilized the ViT structure in various domains, including esophageal endoscopic detection, pneumonia detection, MRI imaging, skin cancer detection, and tumor classification [44,45,46,47]. It is evident that image subjects amenable to ViT application are typically characterized by high spatial information correlation in sequence form, making them well suited for the incorporation of transformer modules such as the self-attention mechanism and position encoding, as shown in Figure 4.

In this training setup, we utilize a Vision Transformer model pre-trained with a patch size of 16 and an input image size of 224 pixels. The input images are the ROIs of the optic disc, which are automatically cropped using an OD localization algorithm described in Section 3.2. The dimensions of the ROIs were standardized to 450 × 450 pixels to ensure uniformity. Subsequently, these images were resized to 224 × 224 pixels to meet the input size requirements of the ViT model. The weights of the Vision Transformer module are frozen to prevent gradient updates, thereby maintaining the learned representations. For the classifier module, we designed a sequential neural network head consisting of an initial linear layer with 512 units, followed by an ReLU activation, a dropout layer with a 25% dropout rate, a subsequent linear layer with 256 units, another ReLU activation, a dropout layer with a 50% dropout rate, and a final linear layer outputting two classes. We observed that the model struggled with binary classification using the binary cross-entropy loss function, so we proposed using the cross-entropy loss function instead, which is more suitable for our two-class classification problem. The model was optimized using the SGD optimizer with a learning rate of 0.0001, momentum set to 0.9, and Nesterov momentum enabled. The training achieved optimal convergence within 25 epochs, employing early stopping with a tolerance of 3 epochs and a threshold of 0.03 to prevent overfitting. The dataset was divided into training, validation, and test sets in the proportions of 80%, 10%, and 10%, respectively. This allocation ensures that the model has ample data for learning while retaining a separate and balanced portion for validation to fine-tune hyperparameters and another portion for unbiased evaluation of its performance.

### 3.4. Hyperspectral Fundus Camera System

The hyperspectral system utilized in this study comprises a TRC NW8/8F fundus system integrated with a DSLR camera, which enables color, red-free, and fluorescein angiography, as shown in Figure 5a. The TRC-NW8 is equipped with a 16.2-megapixel camera. Thus, it delivers high-resolution images featuring a 45° field of view. The TRC-NW8F extends these capabilities by incorporating fluorescein angiography imaging. The spectra were acquired by measuring from a 24-color reference, specifically a 24-color checker, utilizing a spectrometer (QE65000, Ocean Optics, Orlando, FL, USA). Simultaneously, the spectrum emitted from the light source employed in the fundus camera was also recorded (Figure 5b).

## 4. Results

### 4.1. Band Selection Based on Spectral Analysis

In Figure 6a,b, the graphs show the intensity of light across the wavelength spectrum for both glaucomatous and normal eyes. The shaded areas likely represent the standard deviation or a confidence interval, indicating the variability of the intensity measurements among different subjects or images. Figure 6c may represent data derived from a specific region of interest within the retina, while Figure 6d could represent a different or a more general region. The intensity patterns observed in the spectral data differentiate between glaucomatous and normal eyes. These patterns reveal distinct spectral signatures that are indicative of glaucoma. However, solely relying on spectral data makes it challenging to distinguish between glaucomatous and normal eyes with high confidence. In addition, a substantial variation in standard deviation amplitude is observed across the wavelength spectrum, particularly within the range of 550 nm to 780 nm. This phenomenon elucidates the spectral regions with a potential for effective exploitation, irrespective of spectral redundancy factors present throughout the entirety of visible hyperspectral spectra. This finding underscores the importance of discerning and leveraging specific spectral bands for optimal analysis and interpretation, despite inherent redundancies within the spectral data.

In Figure 6c,d, t-SNE analyses were conducted on mean spectral data from retinal rim and optic disc regions. The visual inspection of these analyses reveals significant overlap in data distribution, with no discernible clustering patterns. This observation underscores the inherent challenge in employing conventional discrimination algorithms solely reliant on spectral averages. To address this complexity, advanced classification methodologies are imperative. Integration of deep neural networks, alongside judicious data dimensionality reduction via tailored band interval selection, emerges as a promising avenue for overcoming these limitations and enhancing classification accuracy.

In Figure 7a, this panel shows retinal images captured at different wavelengths, ranging from 380 nm to 780 nm, for both glaucomatous (diseased) and normal (healthy) eyes. The images are divided into two sections labeled OD (oculus dexter, or right eye) and OS (oculus sinister, or left eye), which are common terms in ophthalmology. The top half displays images associated with glaucoma, while the bottom half shows normal eyes. The progression from left to right demonstrates how the retina appears at each specified wavelength, with differences in brightness and contrast that may indicate pathological changes in glaucomatous eyes. In Figure 7b, this section focuses on the optic disc area, providing a close-up view of the same wavelength progression for both glaucomatous and normal eyes. These images can reveal changes in the optic nerve head, which are critical for glaucoma diagnosis. These observations may suggest the presence of significant spectral variations across different band intervals.

Determining the most appropriate method for analyzing spectral differences involves careful consideration of various factors. No universally superior method is available; instead, the suitability depends on specific contextual and data characteristics. Key considerations include the nature of the data, such as noise levels, measurement scale, and the significance of variability. The purpose of analysis is also pivotal—whether it involves identifying subtle differences, focusing on large-scale changes, or understanding overall variability. Moreover, the complexity and computational resources required for each method should be considered, given that some approaches are more computationally intensive than others. 

The signal-to-noise ratio (SNR) is computed for each wavelength by dividing the mean spectrum values by their respective standard deviations. The subsequent step involves determining the absolute difference between the SNRs of the two spectra at each wavelength, as per the following Formula (1):(1)$$SNR\;Difference\left(i\right)=\left|\frac{Spectrum1(i)}{STD1(i)}-\frac{Spectrum2(i)}{STD2(i)}\right|$$

The process of identifying peaks in the SNR typically involves the following mathematical approach: If y(x) represents the signal, the peaks are the points $x_{p}$ where ${y(x}_{p})$ is greater than its neighboring points, i.e., $y\left(x_{p}\right)>{y(x}_{p}\pm 1)$, considering discrete signals. For continuous signals, the criteria involve locating points where the first derivative equals zero and the second derivative is negative, specifically $y^{\prime }\left(x_{p}\right)=0$ and $y^{\prime \prime }\left(x_{p}\right)<0$). Once the peaks are identified, their indices are used to extract the corresponding values from the signal. The band intervals chosen for investigation correspond to the peaks found in the SNR difference, as depicted in Table 1.

The SNR difference is illustrated in Figure 8. Notably, the peak SNR difference value (0.720) occurs at 680 nm, followed by 0.405 at 380 nm. Consequently, we determine the selected band range by identifying the intersection of the SNR difference value at 380 nm with the SNR difference line at the wavelength point of 610.16. Beyond this intersection point, the SNR difference values remain consistently higher. Hence, the investigation focuses on the band range between 610 nm and 780 nm.

### 4.2. Performance Evaluation on Different Band Interval

The performance evaluation of two models, RGB and 610–780, reveals significant differences in their effectiveness, as shown in Table 2. The RGB model achieved an accuracy of 0.8473, precision of 0.8698, recall of 0.8464, F1-score of 0.8447, and an AUC-ROC of 0.8463. These metrics indicate that while the RGB model performs reasonably well, there is room for improvement, particularly in its ability to balance precision and recall. In contrast, the 610–780 model demonstrates superior performance across all metrics, with an accuracy of 0.9008, precision of 0.9009, recall of 0.9007, F1-score of 0.9007, and an AUC-ROC of 0.9007. The substantial improvement in these metrics suggests that the 610–780 model is more adept at correctly classifying instances and distinguishing between classes. The higher precision and recall values indicate a more reliable model with fewer false positives and negatives, respectively. Consequently, the 610–780 model’s enhanced performance metrics underscore its robustness and efficacy, making it a more suitable choice for applications requiring high classification accuracy and reliability.

Additionally, the performance for each narrow spectral band was examined to validate the accuracy of the spectral analysis method. Table 3 presents a comparative performance analysis of models using different spectral bands, including RGB and various narrow bands ranging from 380 to 780 nm. The RGB model achieved an accuracy of 0.8473, precision of 0.8698, recall of 0.8464, F1-score of 0.8447, and AUC-ROC of 0.8463, indicating a strong overall performance. The 380–440 nm band model showed similar accuracy at 0.8397, but with higher precision at 0.8793 and lower recall at 0.7846, resulting in an F1-score of 0.8293 and an AUC-ROC of 0.8393. The models using the 440–485 nm and 485–521 nm bands demonstrated lower performance, with accuracies of 0.7634 and 0.7557, respectively, and both had relatively lower recall and F1-scores. The 521–680 nm band model had an accuracy of 0.7710, with high precision at 0.8889 but lower recall at 0.6154, leading to an F1-score of 0.7273. Notably, the 610–780 nm band model outperformed all others significantly, achieving an accuracy, precision, recall, F1-score, and AUC-ROC all at 0.9007 or higher. This suggests that the 610–780 nm band provides the most discriminative features for the classification task, offering the best balance between precision and recall and overall superior performance metrics.

### 4.3. Ablation Study

The purpose of this ablation study involving models Resnet18, Resnet50, EfficientNet-B0, and ViT (ours) is to systematically evaluate and compare the performance of different neural network architectures on a specific classification task. By analyzing these models, the study aims to identify the strengths and weaknesses of each architecture, determine the most effective model in terms of accuracy, precision, recall, F1-score, and AUC-ROC, and understand the impact of different design choices on model performance. This comparative analysis helps in pinpointing the architectural features and configurations that contribute most significantly to improved classification outcomes, thereby guiding future model selection and optimization efforts in similar tasks. As shown in Table 4, the comparative performance of four models—Resnet18, Resnet50, EfficientNet-B0, and ViT (ours)—reveals distinct variations in their classification capabilities. Resnet18 achieved an accuracy of 0.7863, with precision, recall, F1-score, and AUC-ROC values of 0.8776, 0.6615, 0.7544, and 0.7853, respectively, indicating moderate overall performance with a notable discrepancy between precision and recall. Resnet50 showed improved performance, with an accuracy of 0.8321, precision of 0.8772, recall of 0.7692, F1-score of 0.8197, and AUC-ROC of 0.8316, suggesting a more balanced and effective classification capability. EfficientNet-B0, with an accuracy of 0.7634, precision of 0.8148, recall of 0.6769, F1-score of 0.7395, and AUC-ROC of 0.7627, demonstrated the worst performance among the evaluated models. In contrast, the ViT (ours) model significantly outperformed the others, with an accuracy of 0.9008, precision of 0.9009, recall of 0.9007, F1-score of 0.9007, and AUC-ROC of 0.9007. These metrics highlight the ViT model’s superior ability to classify data accurately and consistently, with high reliability across all evaluated performance indicators. This suggests that the ViT model’s architecture and feature extraction capabilities are exceptionally well suited for the classification task at hand.

## 5. Discussions

The classification results across different band ranges indicate that glaucoma detection is most effective at long-wavelength light ranges. This effectiveness is due to several structural and compositional changes in the retina. Glaucoma leads to the thinning of the RNFL, which reduces light scattering at shorter wavelengths, causing a relative increase in longer wavelength reflectance [48,49,50,51]. Additionally, structural changes in the ONH, such as increased cupping and pallor, further alter reflectance properties favoring longer wavelengths [52,53]. Damage to the retinal pigment epithelium (RPE) and choroid also affects light absorption, contributing to this shift [54,55]. These alterations enhance the detection and monitoring of glaucoma through advanced imaging techniques. The 610–780 nm wavelength band showed superior performance with metrics such as accuracy (0.9008), precision (0.9009), recall (0.9007), F1-score (0.9007), and AUC-ROC (0.9007). The signal-to-noise ratio (SNR) difference supports this, with a significant peak at 680 nm (0.720).

## 6. Conclusions

In the contemporary landscape of an aging population, retinal diseases, once perceived as age-related conditions, now exhibit a trend of earlier onset. This shifting demographic necessitates continuous innovation in drug development and medical testing to enhance the detection and treatment of patients. The integration of artificial intelligence into medical imaging, particularly in the context of glaucoma diagnosis, represents a dynamic and evolving field. Glaucoma is a critical focus due to its potential to cause irreversible blindness if left undiagnosed and untreated. This condition arises when increased intraocular pressure damages the optic nerve, a crucial conduit between the eye and the brain. The challenge lies in its often asymptomatic nature during the early stages, which complicates timely diagnosis. However, early detection through advanced imaging technologies and AI can significantly mitigate the risk of severe vision loss by enabling prompt therapeutic interventions.

This study contributes a unique perspective by focusing on the spectral domain methodology and incorporating spatial information identification through the ViT algorithm. The introduced hyperspectral imaging conversion technique enables the transformation of digital fundus images, which enables revealing variations in spectral regions. The subsequent image reconstruction yields hyperspectral representations, and the ViT algorithm is utilized for classification and comparison across the selected spectral bands.

The performance evaluation of models across different spectral bands demonstrates significant variation in their classification capabilities. The 610–780 nm band model exhibited the highest performance across all metrics, with an accuracy, precision, recall, F1-score, and AUC-ROC all around 0.9007, highlighting its superior discriminative power for the given classification task. The RGB model also performed well, with an accuracy of 0.8473 and balanced precision and recall metrics. However, models utilizing the 380–440 nm, 440–485 nm, 485–521 nm, and 521–680 nm bands showed relatively lower performance, particularly in recall, indicating less effectiveness in identifying true positive instances. These results suggest that while the RGB model is robust, the 610–780 nm band provides the most optimal features for classification, validating the efficacy of spectral analysis in improving model performance. This nuanced approach to metric selection underlines the potential of integrating retinal imaging with artificial intelligence. By leveraging hyperspectral imaging—utilizing spectrometer data for lesion analysis—and ophthalmoscope image analysis, this methodology not only aids healthcare professionals in diagnosing eye conditions but also supports the expansion of telemedicine. This can significantly improve healthcare access, bridging the gap between urban and rural areas, and paving the way for a more inclusive health infrastructure.

The novel hyperspectral imaging approach presents certain limitations. Firstly, the algorithm’s sensitivity is significantly influenced by the uniformity of the illumination. In this study, the Gray World white balance algorithm is employed, utilizing a reference such as the 24 Color Checker. This algorithm estimates the scene’s illuminant as the average RGB value across the image. Additionally, for chromatic adaptation, the Bradford method is applied, suggesting that hyperspectral imaging algorithms are capable of managing chromatic aberration and varying lighting conditions. Secondly, the hyperspectral (HS) imaging algorithms require the subject to be stationary, as they do not account for dynamic factors, which could impact the calibration between the camera and the spectrometer.

In summary, this study advances the field of medical imaging by integrating hyperspectral imaging with the ViT algorithm for the early detection and diagnosis of glaucoma. The novel approach of converting digital fundus images into hyperspectral representations and leveraging the ViT algorithm for classification underscores the potential of this technique in capturing subtle spectral variations. Our performance analysis reveals that Model D, focusing on short-wavelength light, excels in diagnostic metrics, thus highlighting the importance of spectral domain analysis in medical imaging. This research not only demonstrates the efficacy of hyperspectral imaging in improving diagnostic accuracy but also underscores the role of AI in transforming retinal disease detection. By addressing both the strengths and limitations of the proposed methodology, we contribute valuable insights that support the expansion of telemedicine and the development of more inclusive healthcare infrastructure. Our findings advocate for continued innovation in medical imaging technologies to meet the evolving challenges posed by an aging population and the early onset of age-related diseases. This work lays the groundwork for future advancements in AI-driven diagnostic tools, promising enhanced patient care and broader healthcare accessibility.

## Figures and Tables

**Figure 1 diagnostics-14-01285-f001:**
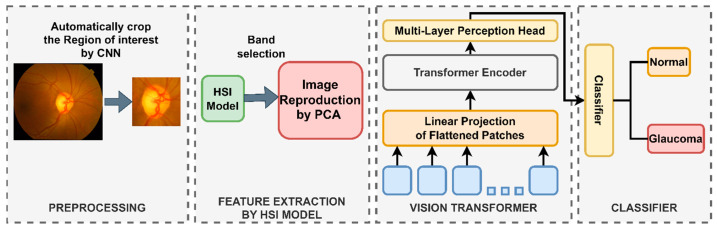
Capturing a novel perspective in glaucoma diagnosis, this study employs a hyperspectral imaging conversion technique to transform digital fundus images. It integrates spatial information identification through a ViT deep learning model. By scrutinizing spectral variations and reconstructing images from specific regions, the research aims to discern disparities between machine learning and conventional clinical approaches. The ViT model then enables classification and comparison across the selected spectral bands, which unveil insights into enhanced diagnostic methodologies for glaucoma.

**Figure 2 diagnostics-14-01285-f002:**
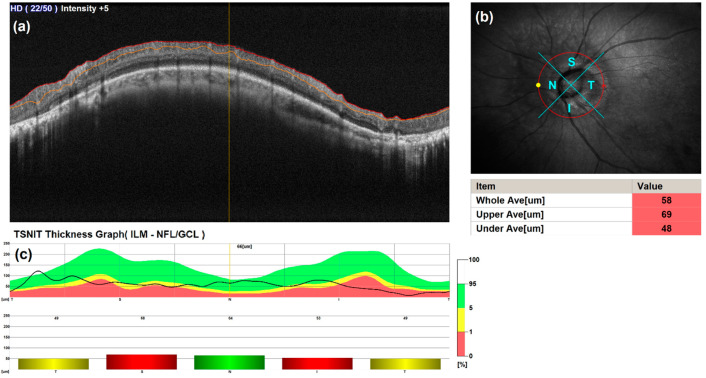
Example of an RNFL report, which includes OCT images, RNFL thickness graphs, and corresponding digital fundus images: (**a**) A circular scan image of the optic disc. (**b**) An enface image for SLO-based eye tracing, displayed from the nasal (N) to the superior (S), to the temporal (T), to the inferior (I) quadrants, based on the cube scan of the optic disc. (**c**) A graph showing the thickness of the RNFL.

**Figure 3 diagnostics-14-01285-f003:**
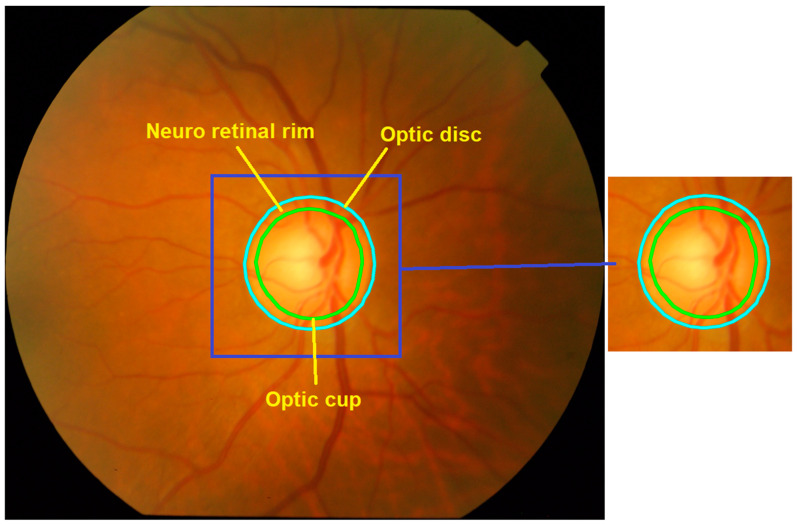
Utilizing the CNN algorithm for automatic cropping of regions containing optic discs and optic cups.

**Figure 4 diagnostics-14-01285-f004:**
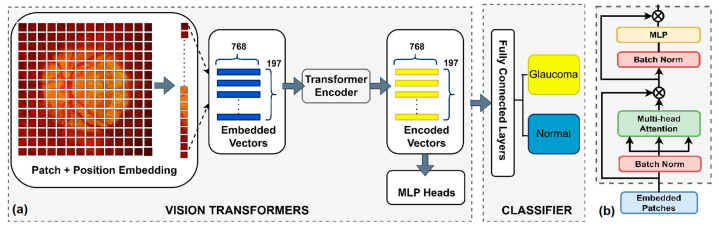
(**a**) ViT inference pipeline. Spatial information and the autocorrelation of feature positions in the fundus image are highly suitable for the self-attention mechanism or embedding position encoding of the transformer encoder module. (**b**) Transformer encoder.

**Figure 5 diagnostics-14-01285-f005:**
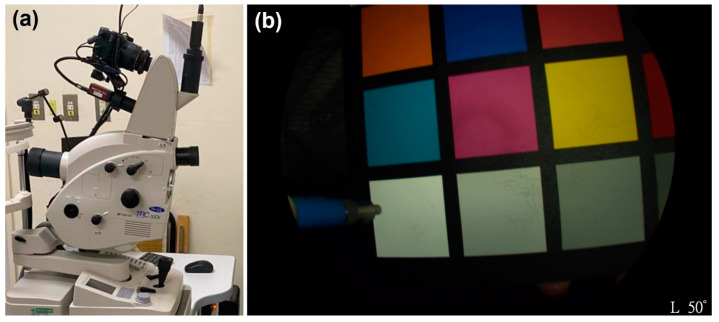
(**a**) Hyperspectral system, which integrates a TRC NW8/8F and a DSLR camera (Canon EOS 90D, Canon Inc., Tokyo, Japan). It enables color, red-free, and fluorescein angiography. (**b**) Spectra were collected using a 24-color checker reference and a spectrometer (Ocean Optics, QE65000) while simultaneously measuring the light source spectrum.

**Figure 6 diagnostics-14-01285-f006:**
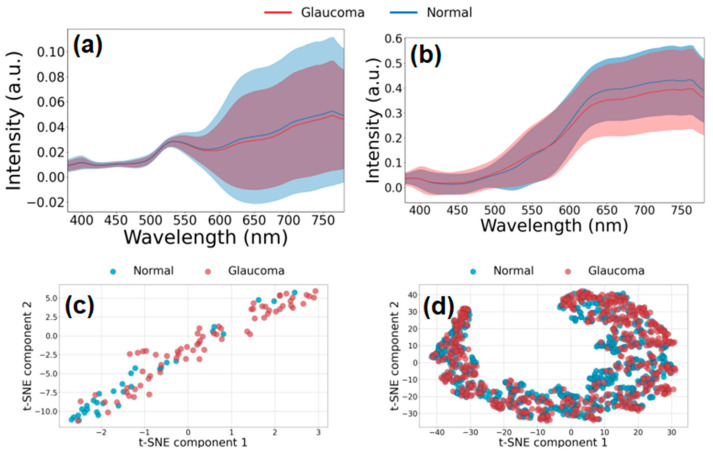
(**a**) The mean reflectance spectra derived from retinal rim areas of glaucoma patients (*n* = 568, depicted in red) and normal participants (*n* = 622, depicted in blue), along with the standard deviation margin, underscore that the observed differences in variations between the two groups are statistically insignificant. Consequently, relying solely on analytical calculations of spectral components proves challenging in discriminating between cases and controls. (**b**) Similarly, the mean reflectance spectra obtained from optic disc areas, which consider glaucoma patients (*n* = 568, represented in red) and normal participants (*n* = 622, represented in blue), yield analogous observations. The t-SNE analysis conducted on mean reflectance spectra data, as depicted in (**c**) for retinal rim areas and (**d**) for optic disc areas, elucidated a lack of distinct clustering patterns between the two groups.

**Figure 7 diagnostics-14-01285-f007:**
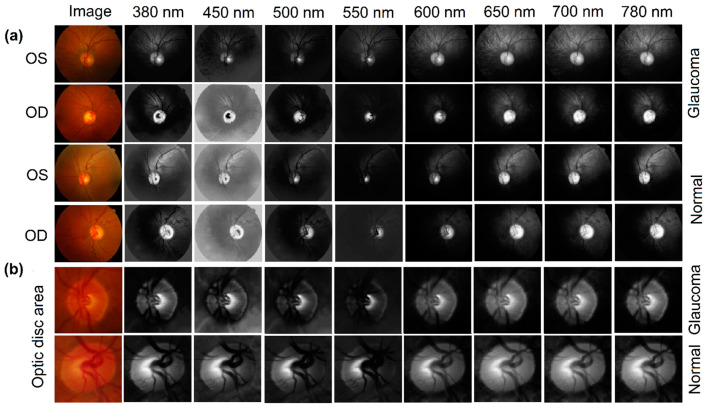
(**a**,**b**) Representative hyperspectral montages of three eyes, which show color-reproduction in range bands from 380 nm to 780 nm, are compared with the original images. The figures in the representative bands range from 380 nm to 780 nm show the variability of retina features.

**Figure 8 diagnostics-14-01285-f008:**
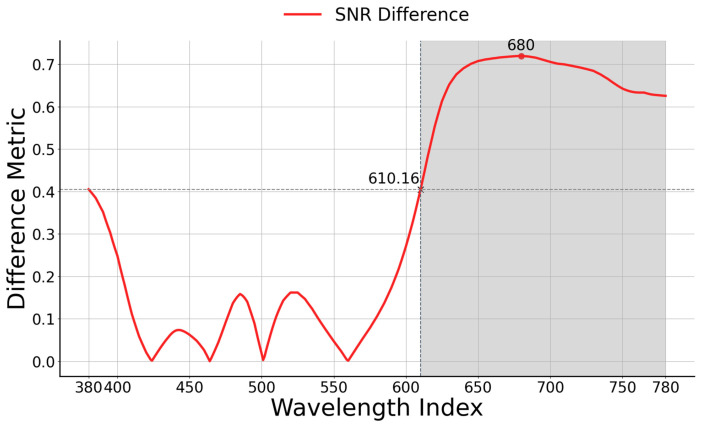
The value of band selection domain in SNR difference analysis. Specifically, the SNR difference value in the region following the wavelength intersection at 610.16 exhibits a higher value compared to the rest of the region.

**Table 1 diagnostics-14-01285-t001:** The SNR difference values at the five peak positions correspond to specific wavelengths.

Wavelength Index (nm)	380	440	485	521	680
SNR difference	0.405	0.074	0.158	0.162	0.720

**Table 2 diagnostics-14-01285-t002:** Performance comparison between fundus images and HS reproduction images from range of 610 to 780 nm.

Dataset	Accuracy	Precision	Recall	F1-Score	AUC-ROC
RGB	0.8473	0.8698	0.8464	0.8447	0.8463
610–780	0.9008	0.9009	0.9007	0.9007	0.9007

**Table 3 diagnostics-14-01285-t003:** Performance comparison between different bands.

Band Interval (nm)	Accuracy	Precision	Recall	F1-Score	AUC-ROC
380–440	0.8397	0.8793	0.7846	0.8293	0.8393
440–485	0.7557	0.8000	0.6769	0.7333	0.7551
485–521	0.7634	0.8148	0.6769	0.7395	0.7627
521–680	0.7710	0.8889	0.6154	0.7273	0.7698
610–780	0.9008	0.9009	0.9007	0.9007	0.9007

**Table 4 diagnostics-14-01285-t004:** Ablation experiments on the 610–780 nm band dataset with different backbone models.

Model	Accuracy	Precision	Recall	F1-Score	AUC-ROC
Resnet18	0.7863	0.8776	0.6615	0.7544	0.7853
Resnet50	0.8321	0.8772	0.7692	0.8197	0.8316
EfficientNet–B0	0.7634	0.8148	0.6769	0.7395	0.7627
ViT (ours)	0.9008	0.9009	0.9007	0.9007	0.9007

## Data Availability

The data presented in this study are available in this article.

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
