# Peer review of "Glaucoma Detection through a Novel Hyperspectral Imaging Band Selection and Vision Transformer Integration"

_diagnostics, 2024, doi:10.3390/diagnostics14121285_

Round 1

Reviewer 1 Report

Comments and Suggestions for Authors

This paper explores the potential of hyperspectral imaging for glaucoma diagnosis. Digital fundus images are transformed into hyperspectral representations, allowing detailed analysis of spectral variations. The Vision Transformer (ViT) model is used for classification and comparison across selected spectral bands. Results show that the 610-780 nm band outperforms others with accuracy, precision, recall, F1-score, and AUC-ROC at 0.9007. The study suggests that hyperspectral imaging, combined with advanced computational techniques, could significantly enhance glaucoma diagnosis. The paper is interesting. However, some issues need to be dealt with, as follows:

1.      The organization of the paper could be improved to enhance readability, particularly in the introduction section and material sections.

2.      The motivation for the study is unclear, the author should outline the main contribution in a separate paragraph.

3.      The author should add a separate section for literature review to better contextualize the research and should be expanded to include recent studies on  glaucoma diagnosis

4.      The experimental setup for the Vit model is not described at all (hyperparameters such as training length, number epochs, optimizer etc. )

5.      The original number of images in glaucoma and Normal classes should be reported.

6.      The split ratio for training, validation, and testing data should be reported.

7.      In line 190       The author mentioned that “demographic data, including age, gender, and underlying health conditions, were collected for each participant”  The author must mention how the research benefited from this data in the diagnosis process and whether it has a role in diagnosis or not If it was used, how was it integrated with the images in the input process to the Vision transformer  model during the training process?

Reviewer 2 Report

Comments and Suggestions for Authors

Novel approach to glaucoma evaluation.

Line 82 - RNFL actually?? Please expand abbreviation here.

Lines 108-140 paragraph -- too long. Please split into at at least two paragraphs.

Line 152 -- New paragraph: "In this study, our aim is to ...."

Line 461 -- Is this line correct? Has the age of dementia onset reduced? Please provide a reference.

Perhaps consider re-writing this line to better explain the importance of glaucoma detection.

Line 501 -- You really need a closing paragraph(s) here summarizing what you have accomplished and your main message to the reader.
